# Sustainability of the Soil Resource in Intensive Production with Organic Contributions

Francia Deyanira Gaytán Martínez [1], Edgar Vladimir Gutiérrez Castorena [1,*], Vicente Vidal Encinia Uribe [1], Rigoberto Eustacio Vázquez Alvarado [1], Emilio Olivares Sáenz [1] and Ma. del Carmen Gutiérrez Castorena [2]

[1] Facultad de Agronomía, Universidad Autónoma de Nuevo León, Francisco I. Madero S/N, Ex. Hacienda el Canadá, General Escobedo 66050, Mexico; francia.gaytanmrt@uanl.edu.mx (F.D.G.M.); vicente.enciniaur@uanl.edu.mx (V.V.E.U.); rigoberto.vazquezal@uanl.edu.mx (R.E.V.A.); emilio.olivaressn@uanl.edu.mx (E.O.S.)

[2] Programa de Edafología, Colegio de Postgraduados, Carretera México-Texcoco km 36.5 Montecillo, Texcoco 56230, Mexico; castor@colpos.mx

* Correspondence: edgar.gutierrezcs@uanl.edu.mx

**Abstract:** Organic agriculture is considered an alternative to sustainably preserve soil fertility. For 10 years, ongoing management with organic contributions was carried out in calcareous soil to support or increase fertility by applying 4 t ha$^{-1}$ of solid poultry manure to produce organic Nopal Verdura (*Opuntia Ficus-Indica*). In addition, during the 2018 cycle, corn was established as an alternative to diversify agricultural production; the crop was monitored by measuring agronomic variables and the normalized differential vegetation index to evaluate the development of different doses of organic inputs with poultry manure, foliar applications with biofertilizers, or in the irrigation system. The soil physical and chemical analysis was carried out from 2015 to 2020 to monitor nitrogen, phosphorus, potassium, calcium, magnesium, and organic matter before planting and after harvest. The results indicated an increase in nitrogen (>50%), calcium (>130%), and magnesium (>20%), while there was a decrease in phosphorus (50%), potassium (60%), and organic matter (18%). The agronomic management caused an increment of EC in the horizon Ap until 12.93 dS m$^{-1}$ at the end of each cycle due to the high ambient temperatures recorded and the inadequate irrigation water quality. We did not find significant differences ($p > 0.05$) in agronomic variables of corn with diverse contributions to organic. However, we obtained a maximum corn yield of 3.9 t ha$^{-1}$ and nopal production of 143 t ha$^{-1}$, despite problems of salinity in the horizons Ap during the agricultural cycle. Overall, processed poultry manure is a sustainable source of macroelements for the production of organic crops in calcisols; however, it is necessary to focus on and counteract potassium depletion and the increase in EC through appropriate agronomic management, with organic contributions, both solid and liquid, to increase or sustain production.

**Keywords:** organic agriculture; calcareous soil; poultry manure; biofertilizers; NDVI

## 1. Introduction

Soil fertility determines the productive potential of agricultural system production. The management techniques influence the physical and chemical properties and crop development. Their effectiveness depends on the characteristics, the quantity and type of residue incorporated, and the type of cultivation proved [1], leading to the search for sustainable practices to improve the content of organic matter and, at the same time, supply the nutritional requirements needed for crops.

Both hemispheres' subtropical arid and semiarid areas have approximately 1 billion hectares of soils classified as Calcisols [2]. México represents 10.4% of the national territory, mainly in the northern states of Sonora, Chihuahua Coahuila, Tamaulipas, and Nuevo León [3]; with important agricultural areas for production, these are generally not suitable

for crop development. However, they are highly productive when these soils are incorporated into agriculture with irrigation and fertilizer. This type of soil is characterized by alkaline reactions and the low availability of organic matter [4] and phosphorus due to the high concentration of Ca and the soil texture [5], which means external sources are required to cover the nutritional needs of crops. Nevertheless, nutritional factors are not only to be considered by producers; the low quality of the underground water irrigation (dissolved salts) can limit the development and yield of the established crop, causing, in turn, the accumulation of dissolved salts in the soil and affect future productions [6].

Organic contributions are an alternative for improving soil fertility and can offer a solution to the issue of water salinity by ensuring better absorption of nutrients for plants, guaranteeing to be a strategy against pollution and environmental deterioration without putting human health at risk by producing food in balance with natural and recyclable resources focused on sustainable agriculture [7,8]. Manure (bovine, goat, ovine, or poultry) is a widely used organic fertilizer that improves the availability of organic material and nutrients, such as nitrogen (N), phosphorus (P), and potassium (K) [9,10] Considering the state of the manure to be used (fresh, processed, leached), the application method (incorporated on the soil, through the irrigation system, foliar application), and the application time (before sowing and according to the established crop stage). Soil environmental factors and the fertilizer's characteristics influence the decomposition and mineralization process [11] and improve the chemical composition and microbial activity [12], thus increasing yield.

Processed poultry manure is commonly used as an organic fertilizer because of its rapid mineralization. It provides essential levels of organic matter (OM) and macronutrients, such as N, P, K, Ca, and Mg, improving the soil structure and the availability of elements to cover the nutritional demand of the plants [13,14]. In addition, the composting process allows the compost to be free from pathogens and weed seeds.

In Mexico, less than 1% of agricultural production is destined to be organic; however, it is one of the countries with the greatest diversity of in-demand crops (81 crops), positioned third place in the world for the number of products and with an area of approximately 500,000 hectares [15,16]. Among the various crops, maize is of the greatest importance among the staple grains worldwide from a food, industrial, political, and social point of view, as well as one of the main crops in Mexico (62% cultivated area) [17], requiring sources with high content of N (22 kg ha$^{-1}$) and K (19 kg ha$^{-1}$) to obtain optimal yields [18]. On the other hand, the nopal vegetable (*Opuntia ficus-Indica* L. Miller) or nopalito is of national importance due to its high consumption. The main producer is in Villa de Milpa Alta in the Southeast of Mexico, with an extension of 68,500 hectares of organic management and an annual production between 90 and 120 t ha$^{-1}$. However, Nuevo León is no exception, being a crop highly adaptable to multiple climatic conditions.

These crops are studied primarily for their variability and genetic diversity that allow them to generate the ability to adapt to different environments, and at the same time create opportunities for agricultural development in the organic sector, obtaining positive yields, and taking precautions with agricultural practices to improve soil sustainability [19].

Technological advances allow for evaluating the nutritional state of N using active optical sensors and light-emitting diodes (LEDs) that are independent of sunlight without the need to destroy plant material that needs to be evaluated [20]; for example, the Green-Seeker TM optical sensor can estimate yield and adequate fertilizer application based on the needs of cereals [21], mainly corn and rice, but the values tend to vary according to factors such as vegetative stage, room temperature, and monitoring time during the day [22].

Moreover, the sensor can quantify the nutritional state in real time using the normalized differential vegetation index (NDVI) [23,24]. The NDVI measures light in the near-infrared (700–1000 nm) and visible red light (620–750 nm) spectra [25] and reports limits values between 0.1 to 0.9 [23] in relation to the energy absorbed (active photosynthetic radiation) by the plant's pigments (chlorophyll) and the energy reflected by cellular structures [26]. Values closer to one mean that the plants are in better nutritional condition,



and this scale allows for creating NDVI thematic maps that distinguish between healthy and sparse or sick vegetation [23].

Our goal was to determine the sustainability of the productive system and soil fertility in an organic-certified farm (from 2009), cultivating nopal vegetable plots that were fertilized with solid processed poultry manure for 10 years in a semiarid region in northeastern Mexico (Nuevo León). We evaluated soil and water quality over five consecutive years (2015–2020), in addition to establishing corn plots and comparing 12 treatments that received poultry manure, organic matter leachate, and biofertilizers during the 2018 cycle for the diversification of organic crops, evaluating and watching the development of the crop using the NDVI (GreenSeeker$^{TM}$).

As an expected result, the application of dehydrated and concentrated poultry manure during the last years in these regions improved soil fertility and solved the salinity issues caused by the underground irrigation water from deep wells. Thus, a sustainable organic agriculture system for cultivating corn and nopal vegetables can be maintained.

## 2. Materials and Methods

### 2.1. Study Area

The study area is located in northeastern Mexico, in the municipality of Zuazua, Nuevo León (25°52′49″ N, 100°05′12″ W). The area has an average temperature of 22.1 °C and receives 548 mm of annual precipitation. The parental material is sedimentary, with the presence of carbonates in the surface and subsurface horizons [6,7]. The farm has an area of four hectares destined for agricultural production; in 2009, it started a program to change from a traditional production system with synthetic fertilization to organic for the crop nopal with processed poultry manure that has been maintained for ten years (2009–2020); in addition, in 2018, it was appropriated for organic corn production, being the only nutritional source used.

### 2.2. Collection of Soil Samples

Soil sampling was carried out for two purposes; the first was to set up the physical and chemical changes between the natural soil and the agricultural soil, through the description of pedological profiles, obtaining altered soil samples for each diagnostic horizon, and the second was to record changes in agricultural soil fertility using composite samples between agricultural cycles.

Sampling was performed over five consecutive agricultural cycles between 2015 and 2020 at the beginning and end of each agricultural cycle. In 2018 exclusively, samples were taken from each treatment, obtaining a composite sample at a depth between 0 and 30 cm. The different methodologies used were obtained from the soil analysis and procedures manual [27]. The samples were dried in the shade at ambient temperature, then ground and sieved with a #30 mesh. The physical and chemical analyses performed were texture (pipette method), wet and dry color (Munsell color chart), apparent density (paraffin), organic carbon (Walkey and Black method), soil reaction (pH) in water (water/soil ratio 1:2), cationic exchange capacity (CEC), interchangeable bases (ammonium acetate methods: Ca and Mg by titration, and N and K by atomic absorption), base saturation from the amount of interchangeable bases, and soluble phosphorus in citric acid. In addition, the descriptions of the site and soil profile were obtained [28].

According to the World Reference Base, both natural and agricultural soils were classified [29]. At the same time, the moisture and temperature regimes for the control section were estimated from the database of the climate station of Marín, Nuevo León (National Weather Service [MX]), and the Newhall simulation model [30].

### 2.3. Analysis of Poultry Manure, Water and Biofertilizer (Not Commercial)

Organic material (poultry manure) used as fertilizer was obtained from poultry farms. The manure was dehydrated and processed (82 °C), ground, sieved, and homogenized (Figure 1). The chemical analyses performed were as follows: pH and electrical conductivity (EC) (NOM-FF-109-SCFI-2007), total nitrogen (Dumas), phosphorus, potassium, calcium, magnesium, sodium, iron, copper, manganese, zinc, and boron (Microwave digestion/ICP), sulfur (Microwave digestion/turbidimetry), humidity (gravimetric method), organic material, ashes, organic carbon (calcination), C/N ratio (dry base), and arsenic, cadmium, chromium, lead, and mercury (ICP-AES: atomic emission spectrometry).

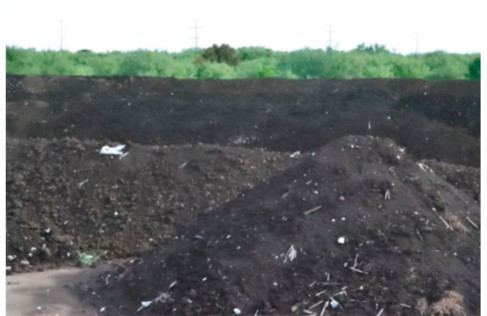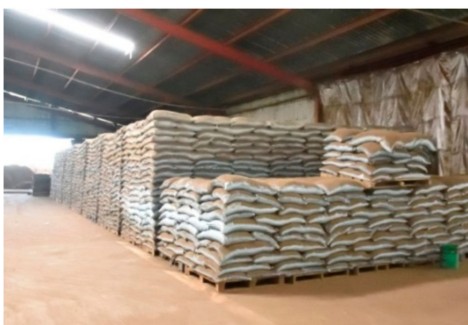

**Figure 1.** Poultry manure processing and storage.

Poultry manure leachate was obtained by fermenting 40 kg of solid manure in 160 L of water for three days. The biofertilizer (local name—Activa-Planta and Activa-Suelo, not commercial) are humic acids of composted manure complemented with potassium hydroxide, seaweed, lignosulfonates, and amino acids.

Water analysis was performed by emission spectrometry and plasma induction (Agilent 725 Series ICP-OES). We obtained the concentration of carbonates, bicarbonates, chlorides, sulfates, calcium, magnesium, sodium, and potassium [31].

### 2.4. Treatments with Organic Contributions (Solid and Liquid)

Organic management was implemented as the only nutritional source on the production of Organic Vegetable Nopal (*Opuntia Ficus-Indica* L. Miller). Each year, at the end of October, the soil is prepared with two steps of disc plowing to incorporate weeds and a third step to incorporate 4 t ha$^{-1}$ of poultry manure. The cladodes planting was established between November and December, with a second application of 1.6 kg m$^2$ of manure. The drip irrigation system was applied 1 to 3 times a week for 12 h. Pest and disease control was carried out through weekly pruning and application of organic insecticides based on neem oils, garlic, and cinnamon monthly, while weed control was carried out manually. Harvesting began at the end of March when the cladodes were 20 to 30 cm long (Figure 2).

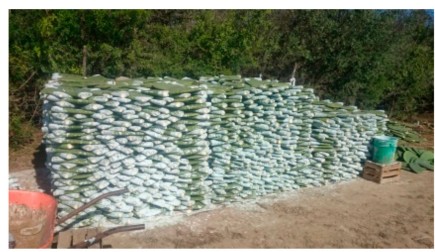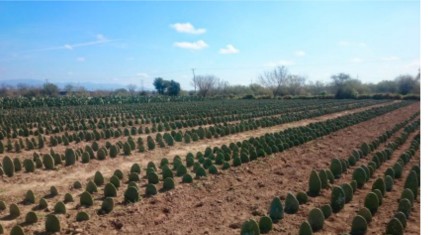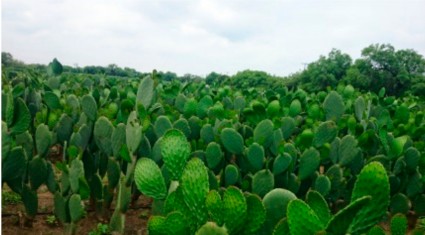

**Figure 2.** Planting and harvesting preparation of organic vegetable nopal.

On the other hand, in 2018, 12 experimental plots were established on an area totaling 4000 m$^2$ to produce corn in a split-plot experimental design; continuing with agronomic management, the plots received 4 t ha$^{-1}$ of solid organic matter (processed poultry manure)

at the beginning of the agricultural cycle, plus the application of biofertilizers during the growing season of the crop through the irrigation system and foliar applications.

Four treatments are described below: plot treatment "A" (TA) and treatment "B" (TB) received 160 L/80 m$^2$ of organic matter leachate and 6.0 L ha$^{-1}$ of Activa-Suelo, respectively; Treatment "C" (TC) received a second application of 4 t ha$^{-1}$ of solid poultry manure, carried out with agricultural machinery, and treatment "D" (TD) was used as a control (no fertilizer application during the growing season). The foliar biofertilization program consisted of applying doses of 1.4 L ha$^{-1}$ (Fd1) and 2.9 L ha$^{-1}$ (Fd2) with Activa-Planta, and a control (Fd0) (Table 1); each experimental unit covered a unit area of 46.67 m$^2$ (with nine replications).

**Table 1.** Organic fertilization in different treatments during standing cultivation.

| Nomenclature | Area Experimental (m$^2$) /9 Replications (m$^2$) | Organic Application at Sowing | + | Organic Contribution during Growing Season | + | Foliar Application with AP® |
|---|---|---|---|---|---|---|
| TA/Fd0 | 93/160 | | | | | NA |
| TA/Fd1 | 46.67/420 | | | 160 L 80 m$^2$ OML | | 1.4 L ha$^{-1}$ |
| TA/Fd2 | 46.67/420 | | | | | 2.9 L ha$^{-1}$ |
| TB/Fd0 | 93/160 | | | | | NA |
| TB/Fd1 | 46.67/420 | | | 6.0 L ha$^{-1}$ de AS® | | 1.4 L ha$^{-1}$ |
| TB/Fd2 | 46.67/420 | 4 t ha$^{-1}$ of PM | | | | 2.9 L ha$^{-1}$ |
| TC/Fd0 | 93/160 | | | | | NA |
| TC/Fd1 | 46.67/420 | | | 4 t ha$^{-1}$ de PM | | 1.4 L ha$^{-1}$ |
| TC/Fd2 | 46.67/420 | | | | | 2.9 L ha$^{-1}$ |
| TD/Fd0$^C$ | 93/160 | | | | | NA |
| TD/Fd1 | 46.67/420 | | | NA | | 1.4 L ha$^{-1}$ |
| TD/Fd2 | 46.67/420 | | | | | 2.9 L ha$^{-1}$ |

TA, TB, TC, TD: treatments; Fd0, Fd1, Fd2: foliar dose of biofertilizer; PM: poultry manure solid, OML: organic matter leachate; AP: Activa-planta; AS: Activa-suelo; NA: not applied; C: control.

The conventional tillage system consisted of (a) moldboard, (b) harrow plow, and (c) furrower. The sowing was carried out with Hualahuises white corn seeds at a total density of 20,000 plants with a separation of 20 cm between them (500 seeds in 100 linear meters) (Figure 3).

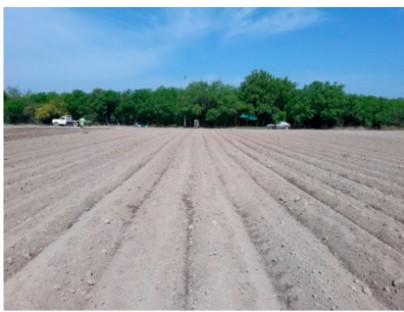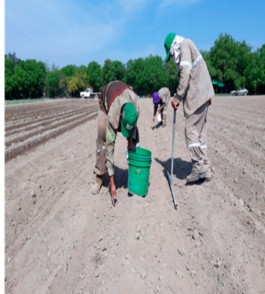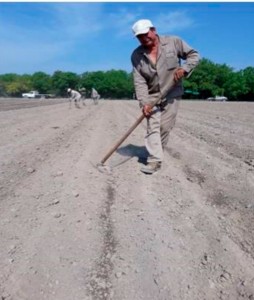

**Figure 3.** Corn sowing and poultry manure incorporation.

*2.5. Soil and Plant Temperature and Normalized Differentiated Vegetation Index*

Leaf and soil surface temperatures were measured directly in the field using an infrared thermometer, while the NDVI was measured with an optical sensor (GreenSeeker$^{TM}$) (Model HCS-100, 2012, Trimble Inc., Sunnyvale, CA, USA). The experimental area and its repetitions were in the field to make a punctual systematic temperature sampling, whereas the NDVI was a valor average of the experimental area, with a separation of 1 m between them. The database registration began in the vegetation stage V3, registered every 15 days until the plants reached physiological maturity.

The data were analyzed, processed, and presented in thematic maps using the Kriging method to compare the temperature and chlorophyll content between treatments. The reclassification of the NDVI values was carried out with amplitudes of 0.20 from the maximum value obtained. The colors assigned in the thematic maps were red (0.61–0.80), yellow (0.41–0.60), green (0.21–0.40), and blue (0.10–0.20) [32,33].

### 2.6. Plant Sampling and Statical Analysis

Plants were collected from each experimental plot 109 days after sowing. Plants were selected at random by height (small, medium, large) due to the variability of these caused by stress to the salinity of the water (Figure 4) through systematic sampling in the experimental plots and treatments. The measured agronomic variables were plant height, leaf area index, weight and length of the stem and leaf, the weight of the corn spike, and dry matter. Corn was harvested 139 days after sowing; we collected and shelled the cob manually from every treatment, thus obtaining the yield. Analyses of variance were performed using the agronomic variables at a level of significance of *p* < 0.05, and the Scheffe test was used to compare means (95%).

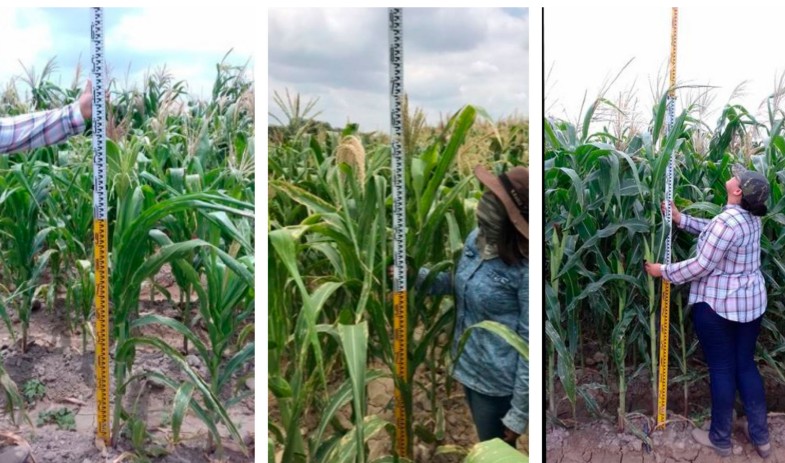

**Figure 4.** Selection of plants of the average height of the experimental unit by treatment.

## 3. Results

### 3.1. Soil Classification of Natural and Agricultural Soil

The natural site reports the native vegetation is predominantly mesquite (*Prosopis glandulosa*), with the presence of cacti, Palo blanco (*Picconia excelsa*) and huizache (*Vachellia farnesiana*); elevation 357 m above mean sea level; regular flat relief with 1% slope; drainage from the normal site without any apparent erosion; sedimentary parental material; presence of mesofauna with worms; use of grazing land occasionally with goats; weather conditions sunny day with light and isolated rains in previous days. The soil presented a superficial horizon A (0–8 cm) with high organic matter content due to the accumulation of litter (mulch). On the other hand, 2BK1, 2BK2, and 2Bk3 horizons (between 8 to 120 cm of the depth) presented slight modifications in their physical characteristics; we collected a composite sample between all the subhorizons for the respective chemical analyses. The natural soil was classified as a Haplic Calcisol (*Siltic*) (Figure 5 and Table 2).

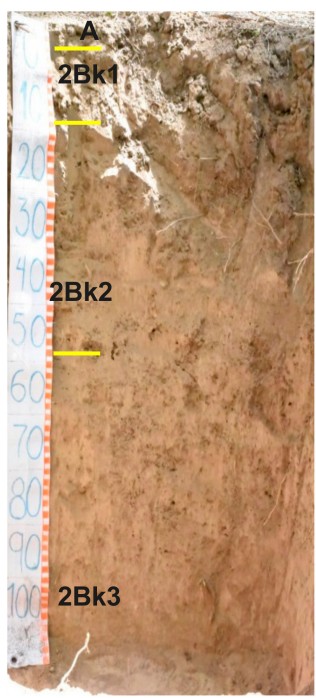

**A** (0–8): Brown (7.5YR 4/3) dry to dark brown (10YR 3/3) moist; silty loam; strong subangular block; slightly hard (dry), and very friable (wet); not plastic; moderate permeability; the presence of crust cracks; frequent fine pores; common thin roots; slight reaction to HCl (slightly calcareous); faint and smooth boundary.

**2Bk1** (8–20): Color brown (10YR 5/3) when dry and dark brown (10YR 3/3) when wet; silty clay loam texture; subangular block structure, strongly developed; very hard when dry and very friable when wet; not plastic; moderate permeability; the presence of interpedal cracks; few coarse pores; few middle roots; with a strong reaction to HCl (highly calcareous); faint horizontal transition.

**2Bk2** (20–60/65): Pale brown (10YR 6/3) dry to brown (10YR5/3) moist; silty clay loam; weak subangular block; hard (dry) and friable (wet); not plastic; moderate permeability; few medium pores; few middle roots; strong reaction to HCl (highly calcareous); faint and wavy boundary.

**2Bk3** (60/65–120): Very pale brown (10YR 7/3) dry to yellowish brown (10YR 5/4) moist; silty clay loam; weak subangular block; slightly hard when dry and friable when wet; not plastic; moderate permeability; few coarse pores; strong reaction to HCl (highly calcareous).

**Figure 5.** Natural soil profile under *Prosopis glandulosa* vegetation.

**Table 2.** Soil natural profile analysis.

| Horizon | Depth | Soil Texture | | | Soil Textural Classes | Bulk Density | Munsell Color | | pH | N |
|---|---|---|---|---|---|---|---|---|---|---|
| | | Sand | Silt | Clay | | | | | H$_2$O | |
| | (cm) | (%) | | | | (g cm$^{-3}$) | Dry | Moist | 2:1 | (ppm) |
| A | 0–8 | 9.7 | 73.6 | 16.8 | Silt loam | 0.82 | 7.5YR4/3 | 10YR3/3 | 6.9 | 1.123 |
| 2Bk1 2Bk2 | 8–60 | 4.6 | 62.8 | 32.6 | Silty clay loam | 1.40 | 10YR 5/3 | 10YR3/3 | 7.7 | 0.116 |

| Horizon | OC | OM | CaCO$_3$ | P | CEC | Exchangeable Cations | | | | PBS |
|---|---|---|---|---|---|---|---|---|---|---|
| | | | | Olsen | NH$_4$Oc, 1N$^{-7}$ | (cmol$^{(+)}$ kg$^{-1}$) | | | | |
| | (%) | | | (Mg/kg$^{-1}$) | (cmol$^{(+)}$ kg$^{-1}$) | Ca$^{++}$ | Mg$^{++}$ | K$^+$ | Na$^+$ | % |
| A | 10.45 | 18.02 | 16.4 | 130 | 32.26 | 11.43 | 2.72 | 1.14 | 0.46 | 48.82 |
| 2Bk1 2Bk2 | 0.75 | 1.29 | 20.7 | 38.5 | 17.66 | 3.75 | 1.01 | 0.83 | 0.39 | 33.86 |

OC: organic carbon; OM: organic matter, PBS: percentage of base saturation.

An aridic moisture regime characterized the agricultural soil (Figures 6 and 7); however, it changed to the ustic regime because they were drip irrigated continuously to conserve their moisture; meanwhile, the temperature regime was hyperthermic. The horizon presented a calcic epipedon (horizon "Ap" or arable layer) with slightly lighter colors and consistency hard and friable in dry and wet conditions respectively, between 0 and 20 cm in depth. The subsurface horizons (Bwk) were eutric and silitic, considered highly calcareous because of the accumulation of secondary CaCO$_3$ and violent reaction to HCl between 20 and 120 cm in depth (Figure 7). Soil texture was silty-loamy with a bulk density that decreased with depth from 1.47 g cm$^{-3}$ to 1.39 g cm$^{-3}$ in the first 89 cm in depth and increased to 1.53 g cm$^{-3}$ in the Bk horizon (89–120 cm) (Table 3).

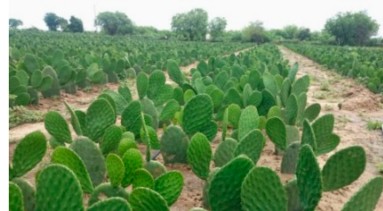 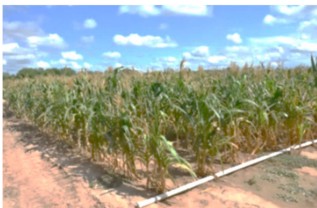

**Figure 6.** The plot of organic agricultural soil with nopal and corn production.

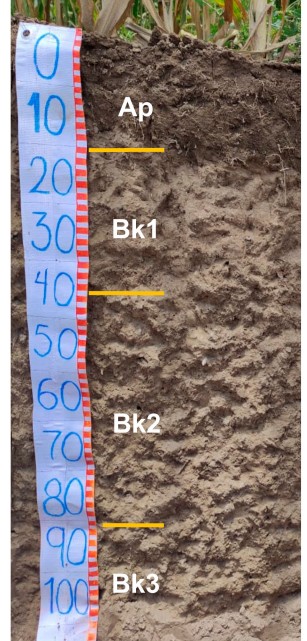

**Ap** (0–20): Pale brownish gray (2.5Y6/2) dry to brownish yellow (10YR5/4) moist; silty-loamy; strongly coarse to fine subangular block, friable, common, fine tubular pores, and few fine fissures; frequent, fine to medium roots; violent reaction to HCL (highly calcareous); wavy and clear boundary.

**Bk1** (20–38/44): Light brownish yellow (2.5Y6/3) dry to dark brownish yellow (10YR4/4) moist; silty-loamy; strongly, coarse to fine subangular block, hard; few tubular pores, and very fine fissures; common, very fine roots and few fine and medium roots; violent reaction to HCL (highly calcareous); broken and gradual boundary. Frequent calcium concretions <1 mm. Common carbon artifacts, 5–10 mm.

**Bk2** (38/44–89): Light brown (2.5Y7/3) dry and brownish yellow (10YR5/4) moist; silty-loamy; strongly very coarse to fine subangular block, very friable; frequent, fine tubular pores; few very fine and fine roots; violent reaction to HCL (highly calcareous); smoot and diffuse boundary.

**Bk3** (89–120): Light brown (2.5Y7/3) dry to brownish yellow (10YR5/4) most; loamy; strongly, very coarse to fine subangular block structure; friable, few tubular pores; few, very fine roots; violent reaction to HCL (highly calcareous).

**Figure 7.** Profile of organic agricultural soil.

**Table 3.** Agricultural soil profile analysis.

| Horizon | Depth (cm) | Soil Texture | | | Soil Textural Classes | Bulk Density (g-cm$^{-3}$) | Munsell Color | | pH H$_2$O 2:01 | N (ppm) |
| | | Sand | Silt | Clay | | | Dry | Moist | | |
| | | (%) | | | | | | | | |
| Ap | 0—20 | 9 | 65.3 | 25.8 | Silt loam | 1.47 | 2.5Y6/2 | 10YR5/4 | 8.2 | 0.102 |
| Bwk | 20–38/44 | 10.2 | 65.6 | 23.9 | Silt loam | 1.43 | 2.5Y6/3 | 10YR4/4 | 8.2 | 0.082 |
| Bk$_1$ | 38/44–89 | 7.6 | 67.8 | 24.5 | Silt loam | 1.39 | 2.5Y7/3 | 10YR5/4 | 8.1 | 0.054 |
| Bk$_2$ | 89–120 | 28.6 | 47.4 | 23.9 | Loam | 1.53 | 2.5Y7/3 | 10YR5/4 | 8 | 0.034 |

| Horizon | OC | OM | CaCO$_3$ | P Olsen | CEC NH$_4$Oc, $1N^{-7}$ | Exchangeable Cations (cmol$^{(+)}$ kg$^{-1}$) | | | | PBS |
| | | | | | | Ca$^{++}$ | Mg$^{++}$ | K$^+$ | Na$^+$ | |
| | (%) | | | (mg/kg$^{-1}$) | (cmol$^{(+)}$ kg$^{-1}$) | | | | | (%) |
| Ap | 1.12 | 1.93 | 18.1 | 58.5 | 13.44 | 6.54 | 1.94 | 0.67 | 0.44 | 71.35 |
| Bwk | 0.75 | 7.29 | 16.4 | 51.7 | 8.64 | 4.61 | 1.06 | 0.66 | 0.44 | 78.36 |
| Bk$_1$ | 0.52 | 0.9 | 16.2 | 16.5 | 12.1 | 3.17 | 1.38 | 0.35 | 0.42 | 43.97 |
| Bk$_2$ | 0.6 | 1.03 | 16.2 | 13.7 | 9.6 | 4.64 | 1.01 | 0.35 | 0.5 | 67.71 |

OC: organic carbon; OM: organic matter; PBS: percentage of base saturation.

The poultry manure facilitated the aggregation of particles in the soil, which presented a strongly developed subangular structure in the Ap (1–20 cm deep) and strongly developed in the Bk1 horizons (20–38/44 cm), developing common channel pores with few fissure pores. Moreover, it facilitated plant development by promoting frequent medium to very fine roots and common fine roots at a depth of 44 cm from the terrain surface; it also presented a friable consistency when wet; however, dry consistency was slightly hard to hard. Thus, the continuous use of poultry manure resulted in cumulative benefits in

agricultural soils compared with natural soils that promote a massive structure with little to no macro or microporosity due to the hard consistency.

Furthermore, they showed that agricultural activity in irrigated soil with underground water caused other pedogenetic processes within the soil profile, such as base saturation of >50% and a cemented or hardened layer between 20 and 100 cm in depth. Therefore, the soils in the study area were classified as Irragric Anthrosols (*Calcic, Eutric, Siltic*) (Figure 7).

### 3.2. Natural and Agricultural Soil Analysis of Fertility

Samples taken at the start of the study (2015) revealed no saline soils to a depth of 120 cm. The natural soil (N.S.) had a high concentration of O.M., N, P, and K in the topsoil (0 to 8 cm) due to the accumulation of leaf litter. Concentrations decreased with increasing depth to 92% for organic matter, 50% for N, 70% for P, and 27% for K in the organic-mineral horizon (8 to 30 cm) with significant increases in salinity, i.e., 183% when compared with the surface horizon (Table 4).

**Table 4.** Chemical analysis of the natural soil and agriculture organic soil profiles (2015).

| Soil | Depth (cm) | pH | EC (dS m$^{-1}$) | O.M. (%) | N (kg ha$^{-1}$) | P | K$^+$ | Ca$^{2+}$ | Mg$^{2+}$ | Na$^+$ |
|------|------------|-----|------|-------|------|------|------|-------|-------|-------|
| | | | | | | | | (cmol$^+$ kg$^{-1}$) | | |
| NS | 0–8 | 6.9 | 0.77 | 10.45 | 140 | 130 | 1.13 | 11.42 | 2.71 | 0.46 |
| | 8–30 | 7.7 | 2.18 | 0.75 | 70 | 38.5 | 0.82 | 3.75 | 1.32 | 0.39 |
| AOS | 0–21 | 8.2 | 0.79 | 1.12 | 62.9 | 58.5 | 0.66 | 6.53 | 1.93 | 0.44 |
| | 21–43 | 8.2 | 0.85 | 0.75 | 51.5 | 51.7 | 0.64 | 4.60 | 1.05 | 0.44 |
| | 43–89 | 8.1 | 0.66 | 0.52 | 69 | 16.5 | 0.34 | 3.16 | 1.13 | 0.42 |
| | 89–120 | 8.0 | 1.20 | 0.60 | 32.2 | 13.7 | 0.34 | 4.63 | 1.00 | 0.50 |

NS: natural soil; AOS: agriculture organic soil; EC: electric conductivity; OM: organic material.

In contrast, the concentration in the surface horizon (0–21 cm) of the AOS (Table 4) was classified as medium for OM, low for N, high for P, and moderate for K; however, in subsurface horizons (up to 120 cm), the concentration of these variables decreased with depth. Ca and Mg were classified as moderately low at the soil surface and low at 21–120 cm depths, while K remained extremely low throughout the profile. The soil profile was alkaline (>8), with an EC <1.0 dS m$^{-1}$ (0–89 cm depths), keeping Na concentrations close to 100 ppm.

### 3.3. Organic Fertilizer (Poultry Manure)

The dehydrated process on poultry manure allowed quality control with little variation in the content and quality of the macro and microelements (Table 5). In comparison between dry and processed manure, the chemical analyses indicated increases in pH, P (20%), K (31%), Ca (41%), and Na (18%). The increment concentrations of salts after dehydrated maintained high levels of EC; nevertheless, it presented a decrease of 11%, as well as in elements N (52%), S (22%), OM (16%), and OC (22%), with N presenting the greatest loss due to the volatilization of $NH_3$ during the compositing process. However, its availability was not affected by a C/N ratio of lower than 20, which allows the rapid decomposition of organic matter by microorganisms. Mg was kept at concentrations of 1.5%. On the other hand, the microelements presented increases in Fe (60%), Cu (368%), Mn (33%), Zn (29%), and B (22%).

**Table 5.** Chemical composition of poultry manure.

| Units | pH | EC (dS m$^{-1}$) | N (%) | P (%) | K (%) | Ca (%) | Mg (%) | Na (%) |
|---|---|---|---|---|---|---|---|---|
| Dry | 6.9 | 19.0 | 4.4 | 2.4 | 2.8 | 8.5 | 1.5 | 0.7 |
| Processed | $7.3 \pm 0.3$ | $16.9 \pm 1.2$ | $2.1 \pm 0.7$ | $2.9 \pm 0.4$ | $3.6 \pm 1.0$ | $12.0 \pm 1.6$ | $1.5 \pm 0.5$ | $0.8 \pm 0.1$ |

| Units | S (ppm) | Fe (ppm) | Mn (ppm) | Zn (ppm) | B (ppm) | OM (%) | OC (%) | C/N |
|---|---|---|---|---|---|---|---|---|
| Dry | 1.60 | 2654 | 486 | 380 | 40.8 | 48.9 | 28.4 | 6.37 |
| Processed | $1.2 \pm 0.3$ | $4265.8 \pm 1497.3$ | $651.2 \pm 92.2$ | $0.8 \pm 77.4$ | $50.0 \pm 10.9$ | $40.9 \pm 2.0$ | $22.0 \pm 3.6$ | $9.2 \pm 1.4$ |

The metals As, Cd, Cr, Pb, and Hg were not detected or were below the method detection limit.

### 3.4. Poultry Manure Leachate

The chemical analysis of Activa-Planta (AP) and Activa-Suelo (AS) (noncommercial) applied as biofertilizers showed a similar concentration of N in both products and high K content (Table 6). Moreover, the analysis demonstrated slightly higher P, Ca, Mg, and Na in AP in comparison with the AS. Other components found in the product were seaweed extract, lignosulfonates, and amino acids, which can increase photosynthesis and microbial activity in the soil and become a food source in areas affected by salt, low organic matter, and soil compaction.

**Table 6.** The chemical concentration of the Activa-Planta and Activa-Suelo biofertilizers.

| Biofertilizers (Noncommercial) | N (%) | P (ppm) | K$^+$ (me L$^{-1}$) | Ca$^{2+}$ (me L$^{-1}$) | Mg$^{2+}$ (me L$^{-1}$) | Na$^+$ (me L$^{-1}$) |
|---|---|---|---|---|---|---|
| Activa-planta | 1.0 | 221.9 | 542.1 | 41.4 | 50.3 | 195.7 |
| Activa-suelo | 1.0 | 66.1 | 300.2 | 22.9 | 38.8 | 117.5 |

### 3.5. Analysis of Irrigation Water (Deep Well)

Irrigation water used for agriculture production was extracted from a deep well classified as "Not Apt" or limited because of its high salt content. The salts in the water become part of the soil solution, so there is a considerable risk of sodification. The sodium absorption ratio (RAS) was 12.8 (Table 7). Moreover, high concentrations of chlorides and sulfates elevate osmotic pressure by decreasing soil humidity, causing osmotic stress, and reducing crop water availability [34]. The groundwater quality can be improved by being pumped into cisterns and passed through organic carbon filters before irrigation. This is suggested by the Centro de Investigación Regional del Noreste (2016). However, the use of this system is limited to producers in the region due to its high cost.

**Table 7.** Chemical characteristics of irrigation water.

| Chemical Analysis | pH | E. C. (dS m$^{-1}$) | CaCO$^3$ (me L$^{-1}$) | HCO$_3^-$ (me L$^{-1}$) | Cl$^-$ (me L$^{-1}$) | SO$_4^{2-}$ (me L$^{-1}$) | Ca$^{2+}$ (me L$^{-1}$) | Mg$^{2+}$ (me L$^{-1}$) | Na$^+$ (me L$^{-1}$) | K$^+$ (me L$^{-1}$) | NO$_3^-$ (me L$^{-1}$) |
|---|---|---|---|---|---|---|---|---|---|---|---|
| Results | 6.5 | 4.43 | 0.00 | 9.28 | 23.2 | 26.44 | 11.48 | 7.71 | 39.72 | 0.01 | 0.81 |

### 3.6. Chemical Analysis of Composite Samples of Soil

The pH of the AOS had a slightly alkaline increment (7 to 8) with an annual application of poultry manure throughout the year in comparison with the NS and a higher electrical conductivity at the end of each agricultural cycle, moving from class C1 (not saline) to C3 (moderately saline) [35]. As a result, the OM of the AOS reported a slight difference in the agriculture cycle 2015 with respect to the NS; however, it decreased from a medium to low

classification [36] over the years, with a total difference of 18% for the last agricultural cycle (2020), although this remained above 1% (Figure 8).

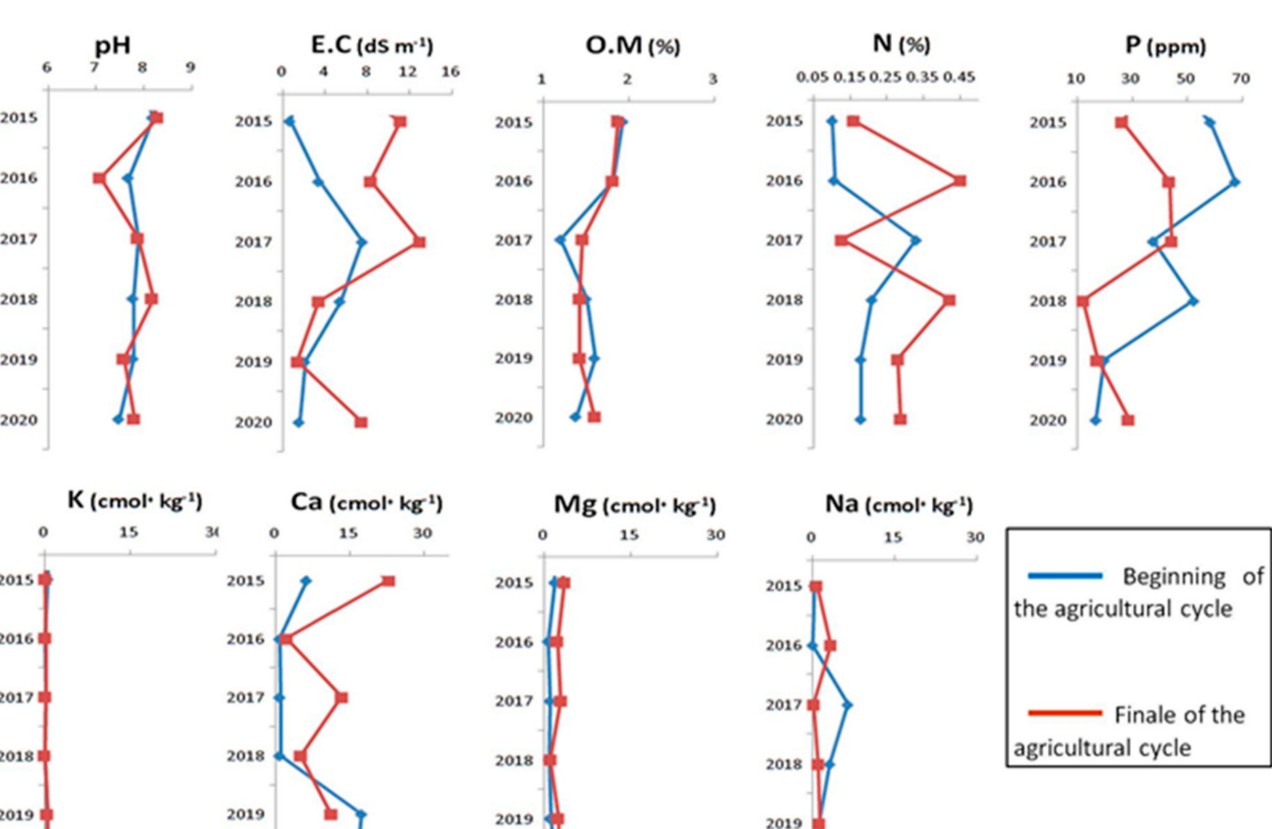

**Figure 8.** Annual variability of chemical characteristics of agricultural soil (0–30 cm).

In most years, the N concentrations increased from 50% to 300% at the end of each agricultural cycle moving from lower or middle class to upper class [37]; nevertheless, they decreased from 26% to 57% during the months without production (winter). On the other hand, P presented decreases in each cycle until reaching a loss of 50%, moving from a very high classification to a high one [38]. Similarity to K decreased by 60% from moderate to very low, according to [39]. Ca and Mg remained at moderate to very high levels, with increases in their concentrations at the end of each cycle: Ca increased between 130% to >1000%, and Mg increased between 21% to >1000%. This increment was due to the fluctuation within the agricultural cycles (the beginning of each period, 2015–2020), which are marked by intense evaporation in summer (June to August) with temperatures of up to 53 °C in bare soil, which increases salts on the soil surface. In winter (November to January), the surface salts decreased due to the rains, decreasing the electrical conductivity in the Ap horizon at the beginning of each agricultural cycle.

### 3.7. Normalized Differential Vegetation Index (NDVI) and Temperatures

During the 2018 agricultural cycle in the crop corn, the NDVI demonstrated values of 0.20 for the V3 stage in all treatments (Figure 9). In addition, as the phenological stages advanced, we observed visual differences and greater vigor with values of 0.60 for V5 stage plants in the TA and TB, whereas the TC and TD plots had values of 0.40. This difference in values between plots continued during the V7 stage with values of 0.80 in the TA and TB plots, which lasted until the VT and R1 stages. In contrast, the TC and TD had an NDVI index of 0.60 until the reproductive stage.

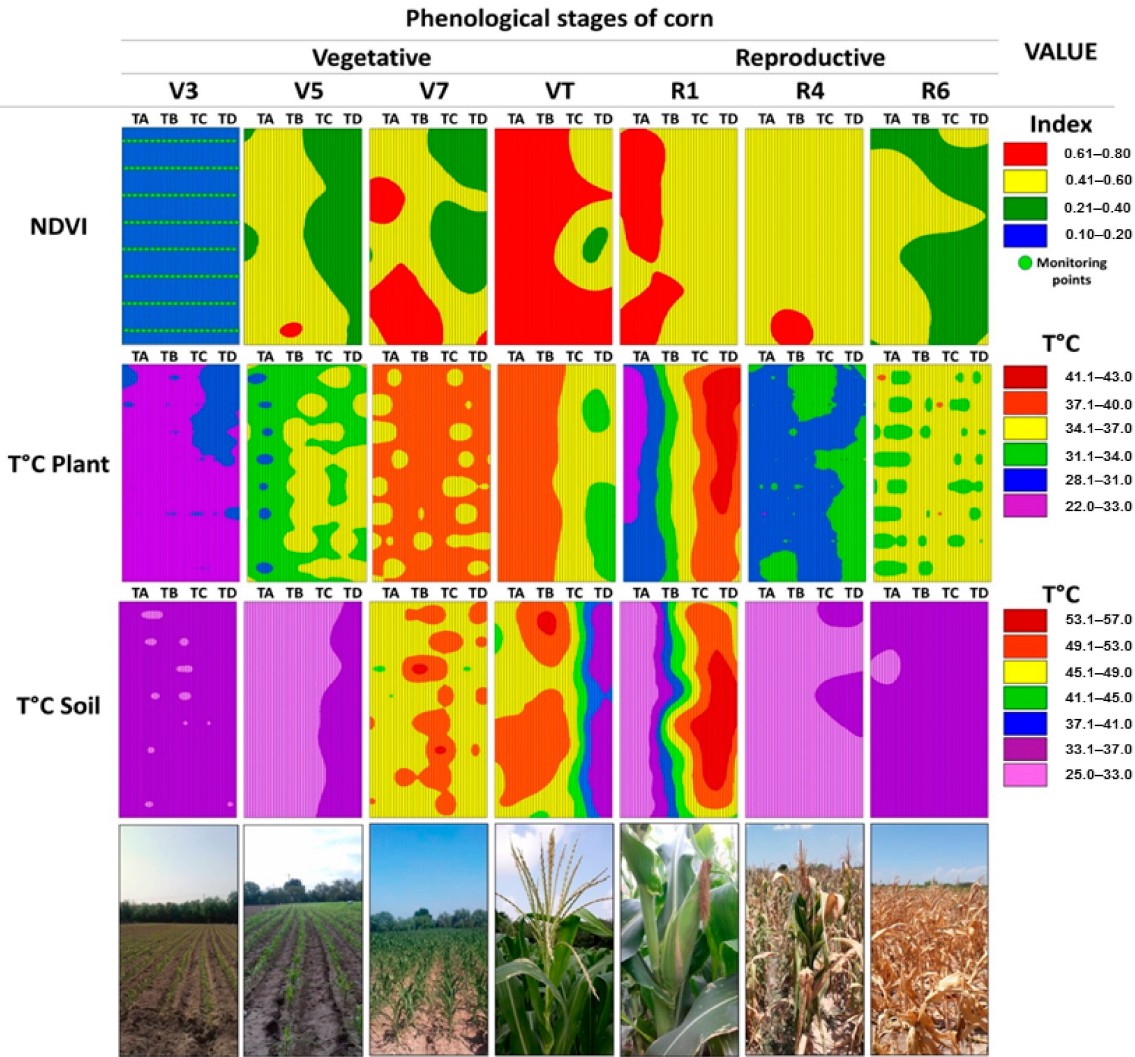

**Figure 9.** Representation of the normalized differential vegetation index (NDVI) values and temperatures for the plants and soils.

At the end of the corn cycle, NDVI values decreased gradually from 0.80 to 0.60 for all experimental plots during the R4 stage and from 0.60 to 0.40 during the R6 stage. The TC and TD plots had values close to zero and plants with signs of water stress and growth inhibition, which affected plant cover and damaged the leaves, decreasing their capacity to absorb and ability to reflect light [40]. In contrast, the TA and TB plots, which received foliar and irrigation biofertilizers, had NDVI values closer to one, showing resistance to salinity stress. However, the thematic maps did not show a visual difference between repetitions and treatments with foliar application.

Soil and plant temperatures increased with the phenological stage, similar to the behavior of the NDVI values. The TA and TB plots reached maximum foliar temperatures of 37.1–40 °C and temperatures of 45.1–53 °C of bare soil in the transition stage (VT). The TC and TD plots had their maximum temperatures (43 °C and 53 °C, respectively) during the R1 stage, which decreased during the crop physiological maturity. The plants in these plots presented water stress and death in the first stages, which affected vegetative vigor, caused delays in growth, and left more soil areas without vegetation cover than the TA and TB plots (fertilization with leachates).

### 3.8. Statistical Analysis

The statistical analysis of sampled plants of corn showed that the incorporation of solid and liquid (leachate) organic sources in the soil did not have a significant effect on plant height and stem length ($p > 0.05$). Despite this, we observed that these values increased as the foliar application doses doubled (Fd2) for each treatment, with a difference of 28% in plant height between the TAFd2 and TDFd0$^C$ (control) treatments. Similarly, the leaf area index did not show statistical differences; however, the value was slightly lower (<4.0) for plants from the TD plot, which only received foliar applications (Figure 10).

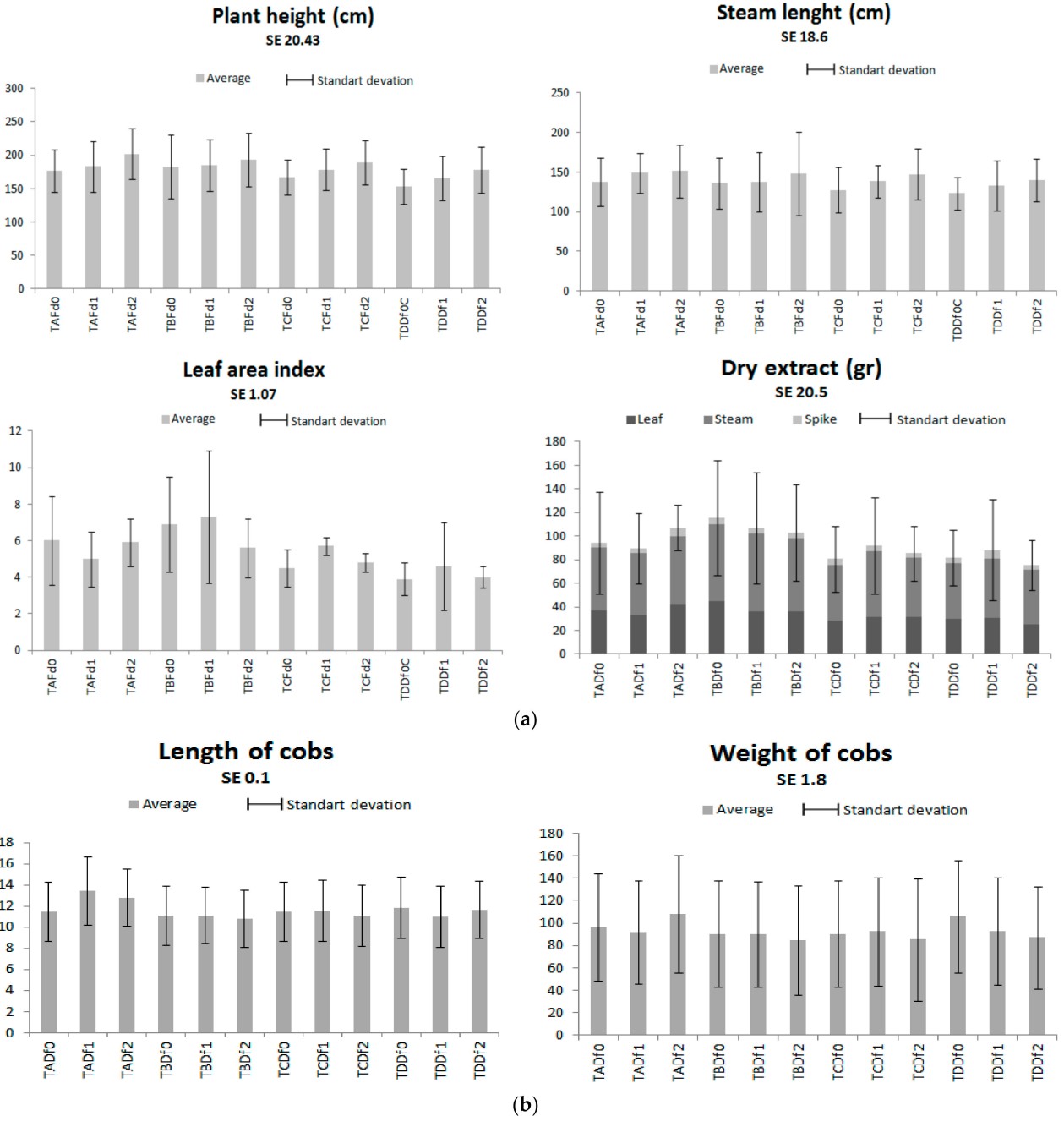

**Figure 10.** Statistical analysis results: plant height, steam length, leaf area index, dey extract (**a**), and length of cobs, weight of cobs (**b**). Statistical analysis (variance analysis and comparison of means) method for organic corn (2018) using the Scheffe (95%); SE: standard error.

The dry matter results were not statistically significant between treatments ($p > 0.05$), with an average difference of 8 g for leaves, 14 g for the stem, and 0.8 g for the spike (Figure 10). An increase in these variables was not observed with the foliar applications doses in each treatment due to the deterioration in the plants in all plots caused by external factors such as pests, disease, the salinity of the water, and the competition for nutrients by native plants, despite organic management with neem extract, garlic extract, pineapple juice fermented with molasses, and manual pruning.

The yield in each plot was calculated by collecting all the corncobs; however, as a result of external factors (mainly stress due to the salinity of the water), cobs were damaged, and losses of between 60% and 70% from a total of 500 plants were recorded independent of foliar applications. These caused statistically significant differences ($p > 0.05$) in the length and weight due to the presence of multiple cobs and incomplete grain filling. In contrast, the plots with leachate fertilization (TA) presented better resistance to water stress, and the highest values in terms of the size and weight of corncobs were obtained (TAFd2) (Figure 10).

Foliar application of Activa-Planta directly affected the yield for every treatment. We recorded a 200% increase for TA, 137% for TB, 300% for TC, and 128% for TD compared with experimental plots without foliar fertilization (Fd0). This increase was due to photosynthetic activity after application with Activa-Planta which also fulfills the need for potassium (Table 6) and thus allows plants to tolerate hydric stress caused by water salinity. The treatment with double foliar applications and poultry manure leachate in irrigation water (TAFd2) led to resistance to this, motivated, in part, by the lack of nutrients at the moment of application, which stimulated natural processes and had a positive effect on the yield (reaching 3.9 t ha$^{-1}$) (Figure 11) [41].

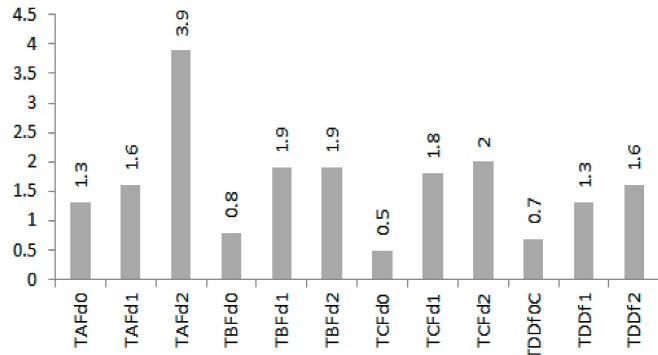

**Figure 11.** Corn yield 2018 per treatments.

While there are salinity problems in the irrigation water for the production of organic corn, it was decided to continue with the production of vegetable nopal, which obtained yields between 63 to 143 t ha$^{-1}$ per year during the agricultural cycle understudy. It should be noted that the nopalito yield was also decreasing (Figure 12).

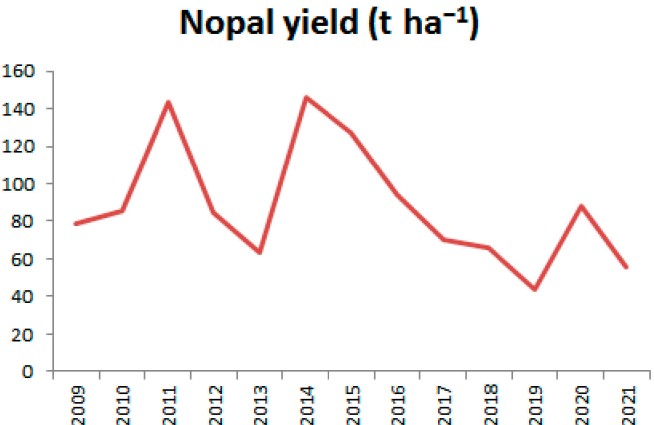

**Figure 12.** Vegetable Nopal yields 2009–2021.

## 4. Discussion

The sustainability of the fertility of soils with contributions of poultry manure was recognized by improving their physical, chemical, and biological properties (macro and microfauna), as well as in yield and quality of different crops; this is due to the increase in the availability of elements [42]. However, these properties vary with climate, manure quality, and soil type [43], as reported in a calcareous soil that increased the concentrations of N, Ca, Mg and Na. Nevertheless, external factors such as low-quality irrigation water and elevated temperatures cause an increase in the EC in the Ap horizon, in turn causing low corn yields into the production of 2018 and a decrease in soil organic matter, phosphorus, and potassium by chemical analysis of 2020.

The annual incorporation of this organic fertilizer to the soil allowed an increase in the activity of macro-organisms in the Ap horizon, mainly earthworms. Consequently, there were more cavities of worms or macropores that helped the movement of water and air within and between the aggregates, improving the permeability of the soil profile compared with natural soil (without agricultural activity). Similarly, the C/N ratio of manure (<20) is considered acceptable [44] for microbial growth so that they act as a regulator between mineralization and nitrogen reserve. In contrast, the climatic conditions of the site (<39 °C) are another factor that favors mineralization [45] at a higher rate, significantly increasing the reserve of elements in the surface and subsurface horizons in the long term [43,46].

At the end of each agricultural cycle, the soil N is used by microorganisms as mineral N ($NH^{+4}$ or $NO^{-3}$), causing the immobilization of this element [47] which causes a decrease in N concentrations at the end of each cycle, requiring the incorporation for the next production cycle and preventing the accumulation of nitrates that are not used after the growth stage of the crop and that become part of the deep waters, as mentioned by Hoover [48].

In the study area, it is common to find increased concentrations of Ca, Mg, and Na in the Ap horizons caused by high evaporation and low precipitation during prolonged periods. Conversely, they decrease from the root zone when the soil receives rainwater. However, agricultural practices, elevated temperatures, and poor well water quality cause the mobility and concentration of soluble salts again in the surface horizon [49], in turn increasing the pH and E.C. [50] and inhibiting the consumption of nutrients for the plant by reducing the water absorption capacity due to saline stress [51]; therefore, the nutrients provided by poultry manure cannot be used by crops, impairing their growth and development during seedling production (height, vegetative vigor, phonological lags, plant death) up to the reproductive stage (flowering, grain filling) [52].

The poultry manure increased these elements in the subsurface horizon after each application and agricultural cycle due to its high contents of a chemical component, mainly calcium (12%) (Table 3) [53,54]. Various papers [48,55,56] have shown that the repeated

application of poultry manure, alone or combined with other natural and chemical sources, in calcareous soils, increases the level of P; however, the high concentration of Ca limited the availability of P by transforming the monobasic ion $H_2PO_4^-$ into the bibasic ion $HPO_4^-$, which reacts with Ca from calcium phosphates, making it insoluble [57,58]; the retention capacity of P in these soils makes it impossible to move to subsurface layers of the profile [59], requiring natural sources that allow the ion to be released and available to the plant.

On the other hand, Taiwo [60] reported other benefits in applying organic fertilizers, reflecting a major reducing K fixation from the soil clays with poultry manure; this allows for improved element solubility and availability to plants. In addition, the release of K allows increasing tolerance to irrigation water salinity, which is essential in the experimental plot. Despite this, the high concentrations of Ca, Mg, and Na cations (Table 5) impair the availability of K by displacing it in the cation exchange complex. Consequently, its absorption by the crop is limited; it causes delays in plant growth and a loss of grains in cereals [61]. Therefore, there is a need to apply a biofertilizer that contains this element to satisfy nutritional requirements and improve yields per unit area.

The low quality of the groundwater (saline water) used by agricultural producers in the region is the main factor that affects the phenological development of the crop; this causes hydric stress during development, variability in height, incomplete grain filling, or death in the early stages [41], as was seen in the TC and TD experimental plots. On the other hand, the application of poultry manure leachate by the irrigation and foliar system reduced the water stress of the plants, promoting vegetative vigor and improving the growth of crops; this shows that it is an effective source of food for plants in agricultural productions with salinity problems [8,62]. In addition, the K concentration in the leachates improved the productivity and quality of the crops by improving the photosynthetic activity supplying a source of proteins, carbohydrates, and fats [63,64].

The increase and decrease in the NDVI values are related to the N concentration and the chlorophyll content in the leaves at each phenological stage (Figure 9). The plots treated with the leachates (TA and TB) presented greater vegetative vigor, exhibiting a higher NDVI value (0.80) than the TC and TD plots, e.g., 0.60 was the maximum value observed during the reproductive stage in plants with growth problems and is associated with poor crop yield [41,65]. This behavior was reflected in the TA and TB plots, which showed the highest vegetation index and, consequently, a higher yield per unit area (3.9 t ha$^{-1}$) than TC and TD plots, which reported lower NDVI values and yields due to reduced plant vigor.

Moreover, temperature is a crucial factor for photosynthetic activity. It is related to the periods of maximum radiation of the plant cover resulting from rapid growth by cell division [66]. This generates increment to temperature on the surface in the leaf during the reproductive stages [67] and heat loss due to evapotranspiration in stressed plants [41].

Despite reports that there is a direct relationship between the reflectance of NDVI values reported by sensor GreenSeeker$^{TM}$, plant temperature (heat emission from leaves), and soil [68], the data from the study area were not statistically significant ($p < 0.05\%$) ($R^2 = 55.2\%$ in plants and $R^2 = 23.9\%$ in soil) among these variables due to external factors that affect their relationships, such as nutritional deficiencies, stress, salinity, drought [69], or crop, as in the case of the nopal vegetable that did not register changes in the values of NDVI at any phenological stage. Nevertheless, the sensor is considered to be an efficient tool to monitor growth [70], detect areas affected by external factors (Figure 9), and predict yield [1,71], as was shown in all plots (Figure 10).

Poultry manure is a soil improver; it can be used in food production, increasing photosynthetic activity [62] and improving yields in adverse situations that limit production. It does this due to its high content of macroelements, mainly N, P, K, Ca, Mg, and organic matter (O.M.) [13,14], and can be seen as a resource in sustainable agriculture. These results are similar to those obtained by Medina [72], who mentioned that the incorporation of organic amendments can improve the chemical properties of the soil, microbial activity, and short-term plant growth. However, the disadvantage is that it does not solve the

salinity problem, limiting the soil fertility and the yields of some crops. As mentioned by Calderón [73], a decrease in the nopal crop yield is caused by the concentration of the EC in the soil, which produces an imbalance in ions, mainly Ca, K and N, causing low availability and absorption of essential elements; in consequence, it is necessary to continue with research focused on the balance between the elements contributed by poultry manure, concentration in the soil, and availability for cultivation in the different vegetative stages.

## 5. Conclusions

Processed poultry manure is a sustainable source of macroelements for the production of organic crops in calcisols by increasing concentrations of N, Ca, and Mg with the application constant between and during the agricultural cycle. In contrast, the availability of P and K is reduced by 50% and 60%, respectively, due to the nature and properties of the soil; however, they can be covered via leachate application of the same product, both on the leaves and through irrigation increased tolerance to the problems caused by salinity by supplying potassium and improving the vigor and vegetative growth of the plants. Our results were ratified using NDVI, which visibly compared crop health through chlorophyll values, obtaining yields into corn of up to 3.9 t ha$^{-1}$ or vegetable nopal of up to 143 t ha$^{-1}$. Nevertheless, the salinity of underground irrigation water limits the increase and the sustainability of the soil resource by maintaining the production system. Therefore, it is necessary to continue investigations in order to establish management and organic contributions, both solid and liquid, to increase or sustain organic agricultural production that the present investigation was unable to determine.

**Author Contributions:** Conceptualization, E.V.G.C. and F.D.G.M.; methodology, E.V.G.C., F.D.G.M. and V.V.E.U.; formal analysis, R.E.V.A., E.O.S. and M.d.C.G.C.; investigation, E.V.G.C. and F.D.G.M.; resources, E.V.G.C., F.D.G.M. and M.d.C.G.C.; writing—original draft preparation, F.D.G.M., E.V.G.C. and V.V.E.U.; writing—review and editing, R.E.V.A., E.O.S. and M.d.C.G.C. All authors have read and agreed to the published version of the manuscript.

**Funding:** This research was found by Programa de Apoyo a la Investigación Científica y Tecnológica (PAICYT/2021) of Universidad Autónoma de Nuevo León (UANL).

**Institutional Review Board Statement:** Not applicable.

**Informed Consent Statement:** Not applicable.

**Data Availability Statement:** Not applicable.

**Acknowledgments:** The authors would like to thank the Programa de Apoyo a la Investigación Científica y Tecnológica (PAICYT) of the Universidad Autónoma de Nuevo León (UANL), the Vertia Company for providing organic products and a plot of land to carry the study and the Registro Nacional de Instituciones y Empresas Científicas y Tecnológicas (No. Registro. 20100247).

**Conflicts of Interest:** The authors declare no conflict of interest. The funders had no rule in the design of the study, in the collection, analyses, or interpretation of data, in the writing of the manuscript, and in the decision to publish the result.

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
