# Peer review of "Sustainability of the Soil Resource in Intensive Production with Organic Contributions"

_agronomy, doi:10.3390/agronomy12010067_

Round 1

Reviewer 1 Report

Row

44 – Calcareous soil – here WRB is cited, but there are not such soil type– it could be Calcisols, Rendzic Leptosols or other – please specify  .

  1. - cationic exchange capacity (C.I.C.) – it is CEC !

241 – Soil profile – if there are carbonates it should be Ap and ABk horizons

246 – Table 2 -  A horizon is 0 – 8 cm ; all other horizons are from 8 – 120 – combined results ? . You show Calcisols without carbonates???

279 – in Morphological description it soil profile is Ak-Bwk, Bk1, Bk2 in table 3 is A, Bk, Bw1 Bw2 – which is real

  1. in Table 3 b PBS – Base Saturation in strongly calcareous soils it should be 100 % -  you do not have ex H or ex Al ??

316 – Table 5 – why you have so high EC – 19 and 16.9 in poultry manure ?

365- table 8 - agricultural soil (0 – 30 cm) and natural soil (NS) – which is agricultural soil  and natural soil . There is on 2 curves in the beginning and in the final  of agricultural cycle – and why EC in the beginning is so high in 2015 and 2017?

Author Response

We appreciate the comments made to the paper and the attention to us. We attach a document where all the concerns made to the document are answered.

Reviewer 2 Report

The manuscript entitled “Sustainability of the soil resource in intensive production with organic contributions” reported poultry manure processed as a sustainable source of soil elements for the production of organic crops in calcareous soils. This manuscript is well written with abundant data and pictures. It would help readers to understanding organic agriculture better. However, there are some questions need to be resolve.

  1. The experiment designs were not clear. How much poultry manure used to crop nopal annually? Line 176: “The cladodes planting is established between November and December, with a second application of 1.6 kg of manure.” It means 1.6 kg ha-1? There was no comparison between synthetic fertilization and poultry manure.
  2. The argument of the manuscript is rather confusing. It was not focused on the subject.
  3. All data in tables and figures have no errors and significance analysis.
  4. It was not easy to understand in figure 8.
  5. There was downtrend with vegetable Nopal yield in figure 11? Please explain the reason.
  6. The discussion of the manuscript is insufficient. There was too much external factors such as the low quality of irrigation water, elevated temperatures cause yield of crop (Figure 10 and Figure 11). How about the synthetic fertilization?
  7. Line 503: The increase and decrease in the NDVI values are related to the N concentration and the chlorophyll content in the leaves at each phenological stage (Figure 8). We can not find the data.

Author Response

(The authors gave the same response as above.)
